# Inhalable Polymeric Nanoparticles for Pulmonary Delivery of Antimicrobial Peptide SET-M33: Antibacterial Activity and Toxicity In Vitro and In Vivo

**DOI:** 10.3390/pharmaceutics15010003

**Published:** 2022-12-20

**Authors:** Laura Cresti, Gemma Conte, Giovanni Cappello, Jlenia Brunetti, Chiara Falciani, Luisa Bracci, Fabiana Quaglia, Francesca Ungaro, Ivana d’Angelo, Alessandro Pini

**Affiliations:** 1Laboratory of Clinical Pathology, Santa Maria alle Scotte University Hospital, 53100 Siena, Italy; 2SetLance srl, 53100 Siena, Italy; 3Department of Medical Biotechnology, University of Siena, 53100 Siena, Italy; 4Department of Environmental, Biological and Pharmaceutical Sciences and Technologies, University of Campania “Luigi Vanvitelli”, 81100 Caserta, Italy; 5Department of Pharmacy, University of Naples Federico II, 80131 Napoli, Italy

**Keywords:** polymeric nanoparticles, lung delivery, inhalable formulations, nanoparticle properties, aerosolization, nanoparticle/mucus interactions, antimicrobial peptides, antimicrobial resistance, *Pseudomonas aeruginosa*

## Abstract

Development of inhalable formulations for delivering peptides to the conductive airways and shielding their interactions with airway barriers, thus enhancing peptide/bacteria interactions, is an important part of peptide-based drug development for lung applications. Here, we report the construction of a biocompatible nanosystem where the antimicrobial peptide SET-M33 is encapsulated within polymeric nanoparticles of poly(lactide-co-glycolide) (PLGA) conjugated with polyethylene glycol (PEG). This system was conceived for better delivery of the peptide to the lungs by aerosol. The encapsulated peptide showed prolonged antibacterial activity, due to its controlled release, and much lower toxicity than the free molecule. The peptide-based nanosystem killed *Pseudomonas aeruginosa* in planktonic and sessile forms in a dose-dependent manner, remaining active up to 72 h after application. The encapsulated peptide showed no cytotoxicity when incubated with human bronchial epithelial cells from healthy individuals and from cystic fibrosis patients, unlike the free peptide, which showed an EC50 of about 22 µM. In vivo acute toxicity studies in experimental animals showed that the peptide nanosystem did not cause any appreciable side effects, and confirmed its ability to mitigate the toxic and lethal effects of free SET-M33.

## 1. Introduction

Antimicrobial resistance is a major public health problem [1]. Multidrug-resistant pathogens are increasing alarmingly, making common infections severe through the development of resistance to antibiotics. There is an urgent need for new antimicrobial drugs and strategies for the treatment of infectious diseases. In the last 20 years, many natural or synthetic antimicrobial peptides have been studied to develop new antibiotics, but few have passed all the preclinical and clinical phases necessary for the transition from a lead compound to a commercial drug [2]. SET-M33 is a non-natural antimicrobial peptide already tested in different infection models in vitro and in vivo. It has a promising efficacy profile for various clinical applications [3,4,5,6,7,8]. Its tetra-branched form confers high resistance to protease and peptidase activities, making these molecules good candidates for in vivo use [9,10,11,12,13,14]. The peptide is active against a panel of clinically prominent Gram-negative bacteria, including many clinical isolates of *Escherichia coli*, *Acinetobacter baumannii*, *Klebsiella pneumoniae*, *Pseudomonas aeruginosa* and other *Enterobacteriaceae* [15,16].

SET-M33 is being assessed for the local treatment of lung infections in cystic fibrosis (CF) patients, where extracellular and cellular airway barriers (i.e., CF mucus, bacteria biofilm) may strongly limit the activity of inhaled antimicrobials. Unfortunately, the physicochemical characteristics of the peptide SET-M33 (a highly charged molecule) do not enable the peptide to easily bypass these biological barriers when administered as a free molecule. Formulation strategies to reach the site of infection are therefore needed.

Among others, a promising formulation approach is encapsulation in a nanometric carrier that can mask the charges of SET-M33, preserve its bioactive structure, reduce side effects, facilitate transport to bacteria and achieve a prolonged therapeutic effect. Excipients currently being assessed for nanocarriers include biodegradable synthetic polyesters, such as poly(lactide-co-glycolide) (PLGA), a promising class of materials for peptide delivery, including by inhalation [17,18,19].

Although PLGA-based nanoparticles (NPs) are considered promising in terms of drug encapsulation, protection, controlled release and aerodynamic properties, these systems do not always efficiently cross extracellular lung barriers. This is the main reason PLGA NPs are often engineered with hydrophilic polymers, such as polyethylene glycol (PEG). PEGylation produces a hydrophilic shell and neutral surface charge that reduce nanoparticle mucoadhesion by avoiding hydrophobic or electrostatic interactions with the mucus gel layer [20]. Nanoparticle muco-inertia can be tuned by changing PEG molecular weight and its density on the particle surface [21,22,23]. In particular, PEG, with a molecular weight in the 2–5 KDa range, provides optimal mucus-penetrating properties [24,25]. Furthermore, particle engineering with PEG is a promising strategy for improving the diffusion of particles across bacterial biofilm, and for promoting the localization of the antimicrobial at the bacterial target [26,27].

This study has two main goals: (1) to design and develop PEGylated PLGA-based NPs for pulmonary delivery of SET-M33; (2) to reduce SET-M33 toxicity when administered directly to the site of infection in the lungs.

## 2. Materials and Methods

### 2.1. Materials

The poly(lactide-co-glycolide) (PLGA) (Resomer^®^ RG 502H uncapped PLGA 50:50, inherent viscosity 0.16–0.24 dL/g) was acquired from Boehringer Ingelheim (Ingelheim, Germany). Poly(lactic-co-glycolic acid) copolymer-Rhodamine-B conjugate (Mn 10,000–30,000 Da) (PLGA-Rhod) was purchased from PolyScitech (Akina division, Inc., West Lafayette, IN, USA). Poly(ethylene glycol) methyl ether-block-poly(lactide-co-glycolide) with PEG average Mn 2000 Da, PLGA Mn 4500 Da and with PEG average Mn 5000 Da, PLGA Mn 7000 Da, diethylenetriaminepentaacetic acid (DTPA), DNA, egg yolk emulsion, poly(L-lysine), RPMI amino acid solution and type II mucin from porcine stomach were purchased from Merck-Sigma Aldrich (St. Louis, MO, USA). Bacterial alginate from the fermentation of *Azotobacter vinelandii* or *Pseudomonas mendocina* was purchased from Carbosynth (UK). Methylene chloride and ethanol were supplied by Carlo Erba (Italy). All salts and reagents were of analytical grade or higher.

### 2.2. Peptide Production

SET-M33 was produced in tetrabranched form by solid-phase synthesis using standard Fmoc (*N*-(9-fluorenyl)methoxycarbonyl) chemistry on Fmoc4-Lys2-Lys-β-Ala Wang resin with a Syro multiple peptide synthesizer (MultiSynTech, Witten, Germany). Side-chain-protecting groups were 2,2,4,6,7-pentamethyldihydrobenzofuran-5-sulfonyl for Arg, *t*-butoxycarbonyl for Lys, and *t*-butyl for Ser. The final product was cleaved from the solid support, deprotected by treatment with TFA containing tri-isopropylsilane and water (95/2.5/2.5), and precipitated with diethyl ether. Crude peptide was purified by reverse-phase chromatography on a column for medium-scale preparation in linear gradient form for 30 min using 0.1% TFA/water as eluent A and methanol as eluent B. The purified peptide was obtained as trifluoroacetate salts (TFacetate). Exchange from TFacetate (toxic by-product) to acetate form was carried out using a quaternary ammonium resin in acetate form (AG1-X8, 100–200 mesh, 1.2 meq/mL capacity). The resin-to-peptide ratio was 2000:1. Resin and peptide were stirred for 1 h, the resin was filtered off and washed extensively, and the peptide was recovered and freeze dried [17]. Final peptide purity and identity were confirmed by reverse-phase chromatography using a Phenomenex Jupiter C18 analytical column (300 Å, 5 mm, 25,064.6 mm) and by mass spectrometry MALDI-TOF/TOF. Rhodamine-labeled peptides were synthesized using Lys-tetramethyl-rhodamine (Lys-TMR) as the first amino acid. HPLC and mass spectrometry characterizations of SET-M33 and Rhodamine-labeled SET-M33 are reported in the Appendix A (Appendix A). 

### 2.3. Quantitative Analysis of SET-M33

SET-M33 was quantified by a RP-HPLC system consisting of a liquid chromatograph (LC-10ADvp), an auto-injector (SIL-10ADvp), a UV–Vis detector (SPD-10Avp) and an integrator (C-R6) (Shimadzu, Japan). The SET-M33 quantitation was performed using a Jupiter 5 µm C18 column (250 × 4.6 mm, 300 Å) (Phenomenex, Torrance, CA, USA) and a mixture of 0.1% (*v/v*) TFA in water and acetonitrile (77:23 *v/v*) as the mobile phase. The flow rate was 1 mL/min and the detection wavelength 215 nm. Calibration curves were achieved by plotting peak area versus the concentration of SET-M33 standard solutions in water, and the linearity of the response was verified over the concentration range 1–200 µg/mL (r^2^ ≥ 0.999).

Rhodamine-labeled SET-M33 was quantified by a spectrofluorimetric assay at λ_ex =_ 520 nm and λ_em_ = 580/640 nm using a plate reader (GloMax^®^ Explorer, Promega, Milano, Italy). The linearity of the response was verified over the SET-M33_Rhod concentration range 0.1–100 µg/mL (r^2^ > 0.999).

### 2.4. Nanoparticle Production

PEGylated PLGA-based NPs were prepared at a SET-M33 theoretical loading of 4% (4 mg of SET-M33 per 100 mg of NPs) by emulsion/solvent diffusion technique, as previously reported [17,28]. Briefly, 100 µL of SET-M33 aqueous solution (0.4 mg) was emulsified with 1 mL of methylene chloride containing a mixture of PLGA (8 mg) and PEG-conjugated PLGA (2 mg) by vortex mixing (2400 min^−1^, Heidolph, Schwabach, Germany). Two formulations were produced employing PLGA conjugated with PEG at different molecular weights (2000 and 5000 Da PEG): M33_PEG2000 NPs and M33_PEG5000 NPs, respectively. The obtained emulsion was added to 12.5 mL of ethanol 96% (*v/v*), allowing polymer precipitation in the form of NPs. The NP dispersion was then diluted with 12.5 mL of water and kept under magnetic stirring for 10 min at room temperature. Afterwards, the organic solvent was evaporated under vacuum at 30 °C (Rotavapor^®^, Heidolph VV 2000, Germany), and the colloidal dispersion of NPs was collected and adjusted with ultrapure water to a final volume of 5 mL. Finally, NPs were isolated by centrifuge at 7000 rcf for 20 min at 4 °C and dispersed in ultrapure water.

Two NP formulations containing fluorescent probes were obtained by encapsulating SET-M33 conjugated with rhodamine B (SET-M33_Rhod) [4] or by embedding PLGA-Rhodamine-B conjugate (PLGA-Rhod) in the polymeric matrix (10% *w/w* with respect to total PLGA amount) (Figure 1).

### 2.5. Nanoparticle Properties

The hydrodynamic diameter (DH), polydispersity index (PDI) and zeta potential (ζ potential) of SET-M33-loaded NPs were measured by dynamic light scattering (DLS) and electrophoretic light scattering (ELS) with a Zetasizer Nano ZS (Malvern Instruments Ltd., Malvern, UK). The ζ potential of the NP aqueous dispersion was evaluated after appropriate dilution in ultrapure water using an electrophoresis cell at a fixed potential of ±150 mV. The reported results are the mean of three measurements for three different batches (*n* = 9) ± SD.

The NP morphology was evaluated by transmission electron microscopy (TEM) using a FEI Tecnai G2 200 kV S-TWIN microscope equipped with a 4 K camera (Thermo Fisher Scientific, Waltham, MA, USA). Briefly, 10 μL of M33-loaded NP dispersion in water (3 mg/mL) was mounted on 200 mesh copper grids coated with carbon film (Ted Pella Inc., Nanovision, Italy) and dried overnight at room temperature.

The actual amount of SET-M33 encapsulated in NPs was evaluated indirectly by measuring the amount of non-encapsulated peptide in the NP dispersion. Briefly, just after production, NPs were collected by centrifugation at 7000 rcf for 20 min at 4 °C (Hettich Zentrifugen, Universal 16R), and the SET-M33 content in the supernatant was analyzed by RP-HPLC, as described above. The results are reported as actual loading (mg of encapsulated SET-M33 per 100 mg of NPs) and encapsulation efficiency (actual loading/theoretical loading × 100) ± SD from three different batches (*n* = 6).

The PEG shell on the NP surface was evaluated by measuring fixed aqueous layer thickness (FALT), i.e., by monitoring the influence of the ionic strength of the dispersing medium on NP surface charge, as previously reported [18,29]. According to the Gouy–Chapmann theory, ζ potential decreases with increasing ionic strength of the dispersion medium. By plotting ln (ζ) against k (k = 3.3 c0.5, where k − 1 is the Debye length), a linear regression is obtained, and the thickness of the fixed aqueous layer (expressed in nm) due to the PEG shell is determined from the slope. Unloaded PLGA-based NPs prepared without PEG were tested in the same conditions as the control.

In vitro release kinetic studies were performed by employing NPs containing SET-M33_Rhod in phosphate buffer at pH 7.2 (120 mM NaCl, 2.7 mM KCl, 10 mM phosphate salts, PBS). SET-M33_Rhod-loaded NPs diluted in PBS (5 mg/mL were incubated at 37 °C and 40 rpm, and in a horizontal shaking water bath (ShakeTemp SW 22, Julabo Italia, Italy). At scheduled time intervals, samples were centrifuged at 7000 rcf for 20 min at 4 °C to isolate NPs, and to withdraw 0.5 mL of the release medium. The released medium was analyzed by spectrofluorometer at λex = 520 nm and λem = 580/640 nm, as described above. Experiments were run in triplicate, and the results are expressed as the percentage of SET-M33 released from NPs (%) ± SD over time.

### 2.6. In Vitro Aerosol Performance of Nanoparticles

Aerosolization properties of SET-M33-loaded NPs were evaluated in vitro on fluorescently labeled NPs prepared employing PLGA-Rhod (i.e., M33-Rhod_PE2000 NPs and M33-Rhod_PEG5000 NPs) after nebulization through an air jet nebulizer (PARI TurboBOY, PARI GmbH, Starnberg, Germany) or a vibrating mesh nebulizer (Aeroneb^®^ Go, Aerogen Ltd., Galway, Ireland). The aerosolization behavior of NPs was evaluated using a next generation impactor (NGI) (Copley Scientific, Nottingham, UK), according to the Comité Européen de Normalization standard methodology for nebulizer systems, with sampling at 15 L/min and insertion of a filter in a micro-orifice collector (MOC). Briefly, the nebulizer reservoir was filled with 1 mL of a water dispersion of SET-M33-loaded fluorescent NPs (1 mg/mL). The nebulizer device was connected to the induction port of the NGI and operated at 15 L/min. The aerosol was drawn through the impactor for 5 min until dry.

The NPs aerosolized (i.e., remaining inside the nebulizer reservoir, deposited on the seven cups of the NGI and in the induction port) were recovered quantitatively in 0.5 N NaOH. The achieved samples were maintained under magnetic stirring at room temperature for 1 h, and the number of fluorescent NPs in the resulting solutions was quantified by spectrofluorimetric analysis at λex = 520 nm and λem = 580–640 nm (GloMax^®^ Explorer, Promega, Italy). Calibration curves were derived by analyzing serial dilutions of a stock of fluorescent NPs after degradation in 0.5 M NaOH, as described above. Each experiment was run in triplicate.

The experimental mass median aerodynamic diameter (MMADexp) and the geometric standard deviation (GSD) were calculated, according to the European Pharmacopoeia, by plotting the cumulative mass of particles retained in each collection cup (expressed as percent of total mass recovered in the impactor) versus the cut-off diameter of the corresponding stage. At the used flow (15 L/min), the cut-off diameters of the NGI stages were 0.98 µm (Cup 1), 1.36 µm (Cup 2), 2.08 µm (Cup 3), 3.3 µm (Cup 4), 5.39 µm (Cup 5), 8.61 µm (Cup 6), 14.1 µm (Cup 7).

The fine particle fraction (FPF) was calculated by considering the number of NPs deposited on stages 3–7 (MMADexp < 5.39 μm) compared to the initial amount loaded in the nebulizer chamber, while the respirable fraction (RF) was determined by the total amount recovered from the NGI.

### 2.7. In Vitro NP Interactions with Mucin and Bacterial Alginates

The interactions of SET-M33-loaded NPs with the main components of mucus and *P. aeruginosa* biofilm extracellular matrix (i.e., mucin and bacterial alginates (BA)) were assessed by turbidimetric measurements, as previously described [18,19]. For mucin/NP interactions, a saturated aqueous solution of mucin was made by dispersing an excess of mucin in water (0.08% *w/v*), under overnight stirring, followed by centrifuging at 6000 rcf and 4 °C for 20 min and collection of the supernatant. For NP/BA interactions, a BA dispersion in water (1% *w/v*) was centrifuged at 6000 rcf and 4 °C for 15 min, and the supernatant was collected. Briefly, 10 µL of SET-M33-loaded NP water dispersion (1 mg) was diluted to 1 mL with the mucin or BA solution and mixed by vortex for 30 s; the turbidity was measured at time 0 and after incubation at 37 °C for 30 and 60 min. The absorbance at an arbitrary unit (650 nm) was recorded by a spectrophotometer. The free SET-M33 in mucin and BA solutions, the mucin and the BA dispersion (without NPs) and the SET-M33-loaded NP dispersions in water were analyzed as controls. Results are expressed as mean absorbance of three replicates at 650 nm ± SD over time. The size of NPs dispersed in mucin and BA was assessed by DLS, as described above. DLS analysis of free SET-M33 in mucin and BA was carried out as a control.

### 2.8. In Vitro Transport of Nanoparticles through Artificial Cystic Fibrosis Mucus and Simulated Bacteria Biofilm

The diffusion of fluorescent NPs through artificial CF mucus (AM) and simulated bacteria biofilm (BA) was evaluated using a previous model based on Transwell^®^ multiwell plates [18,19], with some changes. The AM was prepared as previously described (18), by adding 25 μL sterile egg yolk emulsion, 25 mg mucin, 20 mg DNA, 30 μL aqueous DTPA (1 mg/mL), 25 mg NaCl, 11 mg KCl and 100 μL RPMI 1640 to 5 mL water. The BA was a dispersion of bacterial alginates in water (1% *w/v*), obtained as described previously for NP/BA interactions.

Briefly, 75 μL AM or BA was transferred to Transwells^®^ (6.5 mm; pore size 8 μm; Corning Incorporated, Corning, NY, USA). Then, 25 µL of a NP dispersion in water (20 mg/150 µL) was placed on top of the AM/BA layer, and the wells were inserted into a 24-well plate containing 300 μL acceptor medium per well. Simulated interstitial lung fluid (SILF) and PBS at pH 7.2 were used as acceptor media for AM and BA diffusion experiments, respectively. SILF was prepared as described by Moss [30], and consisted of 600 mg NaCl, 30 mg KCl, 15.7 mg Na_2_HPO_4_, 7.1 mg Na_2_SO_4_, 27.8 mg CaCl_2_, 57.4 mg NaH_3_C_2_O_2_, 260.8 mg NaHCO_3_, 9.8 mg Na_2_H_5_C_6_O_7_*H_2_O and 29.4 mg MgCl_2_*6 H_2_O in 100 mL water. At scheduled time points (from 0 to 6 h), the acceptor medium was sampled and centrifuged at 9000 rcf for 20 min at 4 °C to isolate NPs. The NP pellet was suspended in 50 µL water, diluted with 450 µL 0.5 N NaOH, and stirred for 1 h to achieve complete PLGA matrix degradation. The resulting samples were analyzed by spectrofluorimetric analysis at λex = 520 nm and λem = 580–640 nm (GloMax^®^ Explorer, Promega, Italy) to quantify the number of permeated fluorescent NPs. Calibration curves were obtained by analyzing serial dilutions of SET-M33-loaded fluorescent NP standard solutions prepared from a stock of fluorescent NPs degraded in 0.5 M NaOH. The linearity of the response was verified over the concentration range 10–1000 μg/mL (r2 ≥ 0.999). Experiments were run in triplicate, and the results are expressed as the percentage of total NPs permeating over time ± SD.

The ability of free SET-M33 to diffuse through AM and BA was also evaluated using SET-M33_Rhod. The diffusion test was carried out on free SET-M33_Rhod, and compared to M33-Rhod_PEG2000 NPs and M33-Rhod_PEG5000 NPs. Briefly, 25 μL of a SET-M33_Rhod aqueous solution (4.8 mg/mL), or 25 μL of a M33-Rhod-loaded NP dispersion (20 mg/150 µL corresponding to 4.8 mg/mL of SET-M33_Rhod), was added on top of AM or BA. At scheduled time intervals (from 0 to 6 h), the acceptor medium (i.e., SILF or PBS) was collected and the amount of SET-M33_Rhod permeated was determined by spectrofluorometer, as described above. In the case of SET-M33_Rhod-loaded NPs, the amount of SET-M33 diffused was evaluated after degradation of permeated NPs in NaOH. Experiments were run in triplicate, and the results are expressed as the percentage of permeated SET-M33_Rhod ± SD.

### 2.9. Antibacterial Activity In Vitro

*Pseudomonas aeruginosa* (ATCC 27853™) was grown at 37 °C in Luria Bertani (LB) medium to an optical density (OD) of 0.8 (λ = 590 nm). The bacterial culture was centrifuged at 4000 RPM for 10 min at 4 °C, washed in PBS, and resuspended in minimal medium E supplemented with 0.2% glucose and 1 μg/mL vitamin B1 (medium E++) [31]. The bacterial culture was then diluted to a concentration of 2 × 10^5^ colony forming units (CFU)/mL in medium E++. M33_PEG5000 NPs were solubilized in medium E++ to final peptide concentrations of 24 µM, 12 µM and 6 µM. In a 96-well plate, 100 μL of bacterial culture was added to 100 μL of NPs loaded with SET-M33 peptide at different concentrations; 100 μL of bacterial culture plus 100 μL of medium E++, and 100 μL of bacterial culture plus 100 μL of unloaded NPs, were used as controls. The plate was incubated at 30 °C. After 24, 48 and 72 h, an aliquot from each well was withdrawn and plated in an LB agar plate for the bacterial colony count. Statistically significant differences between groups were evaluated by one-way ANOVA with Dunnett’s multiple comparison test using GraphPad Prism 5.03 software.

### 2.10. Antibiofilm Activity

*Pseudomonas aeruginosa* (ATCC 27853™) biofilm was produced using special microtiter plates with lids bearing 96 pegs (Innovotech Inc., Edmonton, AB, Canada), according to a previous procedure [8,32]. Briefly, 200 µL of the bacterial culture (1 × 10^7^ cells/mL) in LB medium was transferred to each well. Plates were sealed with the peg lids, on which biofilm cells can grow, and then placed in a humidified orbital incubator at 35 °C for 20 h under agitation at 100 rpm. After biofilm formation, the peg lids were rinsed twice with PBS to remove planktonic cells, and transferred to a 96-well microtiter challenge plate, each well containing 200 µL of free SET-M33 (peptide concentrations: 24 µM, 12 µM and 6 µM), M33_PEG5000 NPs (peptide concentrations: 24 µM, 12 µM and 6 µM) and unloaded NPs dissolved in minimal medium E supplemented with 0.2% glucose and 1 µg/mL vitamin B1 (medium E++) [31]. Bacteria in complete medium E++ and bacteria incubated with unloaded NPs were used as controls. The plates were incubated at 30 °C for 24, 48 and 72 h. For biomass evaluation, pegs were then washed twice with PBS and fixed in 4% paraformaldehyde solution in PBS for 15 min at room temperature. Pegs were stained with 0.05% (in water) crystal violet (CV) solution for 15 min. The excess CV was removed by washing the pegs with water. Finally, bound CV was released from pegs using absolute ethanol. The absorbance was measured at 600 nm, and the percentage of biofilm biomass was calculated with respect to the control (100% biofilm biomass). Statistically significant differences between groups were evaluated by one-way ANOVA with Dunnett’s multiple comparison test using GraphPad Prism 5.03 software.

### 2.11. Cytotoxicity

CFBE41o- and 16HBE14o- cells (obtained by Prof. Dieter Gruenert, University of California San Francisco under specific agreement) were plated at a density of 2.5 × 10^4^ per well in 96-well microplates, previously incubated with coating solution (88% LHC basal medium, 10% bovine serum albumin, 30 μg/mL bovine collagen type I and 1% human fibronectin). Free SET-M33 and M33_PEG5000 NPs were diluted in culture medium (Minimum Essential Medium (MEM) with Earle’s salts, 10% FBS, 60 µg/mL penicillin, 100 µg/mL streptomycin and 200 µg/mL glutamine) in order to achieve final SET-M33 concentrations from 100 to 3.125 µM, and added 24 h after plating. Cells were grown for 48 h at 37 °C under 5% CO2. Viability was assessed with 0.1% CV solution. Cells were fixed with PBS–4% paraformaldehyde for 15 min at room temperature and stained with 0.1% CV solution for 1 h at room temperature. The cells were then solubilized with 10% acetic acid, and the absorbance was measured at 595 nm using a microplate reader. EC50 values were calculated by non-linear regression analysis using GraphPad Prism 5.03 software.

### 2.12. In Vivo Toxicity

Fifteen female 20 g BALB/c mice (Charles River) were used for this toxicity experiment. After an acclimatization period of 4 days, the mice were divided as follows: group 1 (*n* = 5 mice; free SET-M33 at 10 mg/Kg), group 2 (*n* = 5 mice; SET-M33-loaded NPs at 10 mg/Kg), group 3 (*n* = 5 mice; unloaded NPs). The mice were first anesthetized intraperitoneally with Zoletil 50/50 (250 mg tielamine + 250 mg zolazepam) + nerfastin (20 mg xilazine). Then they were placed on specific mouse holders and treated with a single intratracheal (i.t.) administration of free SET-M33 (10 mg/Kg) or M33_PEG5000 NPs (263 mg/Kg, corresponding to 10 mg/Kg of SET-M33) or unloaded NPs (263 mg/Kg) at a volume of 25 µL/mouse. The delivery system used for the i.t. administration was a PennCentury™ dry powder insufflator for mice (FMJ-250, Penn-Century Inc., Wyndmoor, PA, USA) [33,34]. The mice were observed for 96 h after inoculation. They were weighed every day from arrival to the last day of the experiment. Moribund animals were killed humanely with 3.5% isoflurane and CO_2_ to avoid unnecessary distress. A toxicity score was assigned as follows: wiry coat and poor motility = mild signs; very wiry coat, abundant lachrymation and poor motility even under stimulation = manifest signs.

## 3. Results

### 3.1. SET-M33-Loaded Nanoparticle Properties

Two different formulations of PEGylated PLGA-based NPs containing SET-M33 were prepared by a modified emulsion/solvent diffusion technique employing PLGA conjugated with PEG 2000 Da or 5000 Da. 

The hydrodynamic diameter (DH), polydispersity index (PDI) and zeta potential (ζ potential) of SET-M33-loaded NPs were determined by DLS and ELS, as described above, and the results are reported in Table 1.

Both NP formulations showed a D_H_ close to 200 nm, sufficiently small to cross lung barriers, negative ζ potential and high EE (≥86%), suggesting that the PEG molecular weight did not significantly affect particle properties.

As confirmed by TEM analysis, both the PEGylated NP formulations showed a regular and spherical shape (Figure 2).

The FALT analysis of the PEG shell on the NPs surface, obtained by monitoring the influence of the ionic strength of the dispersing medium on NP surface charge, confirmed the core–shell structure of the NPs, with a PLGA core surrounded by a PEG shell (Figure 3). This hypothesis was also supported by the FALT evaluation of NPs prepared without PEG, which exhibited a very low shell thickness (1.08 nm).

As expected, particles with PEG 5000 Da showed a higher shell thickness (4.07 nm) than that obtained with PEG 2000 Da (3.21 nm).

The in vitro kinetics of SET-M33 release were evaluated in PBS employing NPs containing the SET-M33_Rhod. The results are shown in Figure 4 as the percentage of peptide released over time.

The two NP formulations showed typical biphasic release profiles characterized by an initial burst (about 60% and 45% of the amount encapsulated released in 6 h for M33-Rhod_PEG5000 NPs and M33-Rhod_PEG2000 NPs, respectively) followed by controlled release of the peptide lasting about 7 days. In the case of polymeric nanoparticles, the diffusion of the drug is generally faster than polymeric matrix erosion, thus the mechanism of release can be predominantly related to the diffusion process.

### 3.2. In Vitro Aerosol Performance of Nanoparticles

The in vitro aerosol performance of NPs is shown in Figure 5 and Table 2.

Cumulative mass recovered as a function of the cut-off diameter and the fine particle characteristics of the aerosol cloud of M33-Rhod_PE2000 NPs and M33-Rhod_PEG5000 NPs was evaluated upon delivery through Aeroneb^®^ Go and PARI TurboBOY nebulizers.

Differences were observed in the deposition patterns of the NP formulations tested. When PARI TurboBOY was used as the nebulizer, both NP formulations generated a NP cloud with a low aerodynamic diameter (˂2.7 µm) that reached the deep CUP of the NGI. On the other hand, when Aeroneb Go was used, M33_PEG5000 NPs showed a higher aerodynamic diameter (3.85 ± 0.70 µm) than M33_PEG2000 NPs (2.68 ± 1.59 µm). Nevertheless, all the aerodynamic diameters were less than 5 µm, suggesting that the distribution of the formulation in the lungs by the most widely used nebulizers is uniform.

### 3.3. In Vitro NP Interactions with Mucin and Bacterial Alginates

A critical issue in defining the ability of particles to diffuse across lung barriers (i.e., the mucus layer) is the evaluation of the interactions between the particles and major barrier components. An absence of interactions is commonly considered a prerequisite for improving particle diffusion across the mucus layer [35]. We therefore first derived a rough estimation of interactions between NPs and mucin, the major component of airway mucus, and between NPs and BA, a main component of the *P. aeruginosa* biofilm extracellular matrix. This was achieved by measuring NP scattering at 650 nm with and without mucin/BA.

As shown in Figure 6A, both formulations showed the same absorbance in water and in mucin, suggesting that mucin is not absorbed on the surface of NPs, irrespective of the length of the PEG chain on the particle surface. The absence of evident particle interactions was also confirmed by DLS size analysis, where NP dispersion in water and mucin was comparable (Figure 6B).

On the other hand, M33_PEG2000 NP dispersion in BA showed a statistically significant difference (*p* ˂ 0.0001) in the scattering, suggesting an interaction between NPs and BA (Figure 6). This effect was not observed with M33_PEG5000 NPs, where the greater thickness of the PEG shell protected the particle surface from these interactions. Although detectable, the slight increase in scattering by PEG 2000 particles suggests an absence of particle aggregation and strong interactions, as confirmed by size analysis in BA (Figure 6B).

On the other hand, free SET-M33 strongly interacted with mucin and BA, leading to the formation of nanometric complexes and 60-fold and 200-fold increases in sample turbidity, respectively (Figure 7), while PEGylated NPs efficiently shielded interactions of SET-M33 with mucin and BA, avoiding any formation of complexes or aggregates.

### 3.4. In Vitro Diffusion Studies across Mucus and Biofilm Models

The ability of NPs to aid the diffusion of SET-M33 through artificial mucus (AM) and simulated bacterial biofilm (BA), and the effect of the PEG shell on NP diffusion, were evaluated by a modified Transwell^®^ multiplate assay (Figure 8).

After 1 h, NP percentages as high as 70% for M33_PEG5000 NPs in the acceptor medium were recorded, demonstrating that these PEGylated NPs readily diffused across the artificial mucus layer (Figure 8A). M33_PEG2000 NPs did not diffuse so readily even after 4 h (Figure 8A). Although the two PEGylated NPs diffused rapidly through the mucus layer, the highest diffusion was observed for NPs modified with PEG 5000 Da, which provides an optimal dense PEG shell with a brush conformation, confirming that the PEG molecular weight affects the ability of NPs to diffuse across the mucus layer [36]. On the other hand, in the BA diffusion studies, no differences were found between the two NP formulations in the first 2 h (see similar diffusion profiles in Figure 8C). An effect of PEG molecular weight was observed towards the end of the assay at 6 h, namely, the greater diffusion of M33_PEG5000 NPs (90%) than of M33_PEG 2000 NPs (70%).

The ability of PEGylated NPs to aid the diffusion of SET-M33 across simulated lung barriers was confirmed by in vitro permeation studies, performed under the same conditions, with NPs containing fluorescently labeled SET-M33. The results were compared with those of free SET-M33_Rhod (Figure 8B,D). After deposition of the same amount on the AM layer, the percentage of SET-M33_Rhod encapsulated in M33_PEG 5000 NPs permeating into the acceptor compartment in 1 h was more than twice that of the free SET-M33_Rhod and SET-M33_Rhod encapsulated in M33_PEG 2000 NPs (80.68% for M33_PEG 5000 NPs versus 37.10% and 26.32% for free SET-M33_Rhod and M33_PEG 2000 NPs, respectively) (Figure 8B). Interestingly, while a slight decrease in SET-M33_Rhod diffusion was detected in AM, high interaction was detected between free SET-M33_Rhod and BA, as suggested by the permeation profile (Figure 8D). The encapsulation of SET-M33_Rhod in PEGylated PLGA NPs led to a 7.6-fold increase in SET-M33_Rhod diffusion across the BA layer. After 6 h, only 12.3% of SET-M33_Rhod had crossed the BA layer, while more than 90% was detected in the acceptor compartment when the peptide was encapsulated in NPs (93.74% and 92.91% for M33_PEG 5000 NPs and M33_PEG 2000 NPs, respectively). These results indicate that PEG5000 NPs were the best formulation. They were therefore selected for further experiments on in vitro antimicrobial and antibiofilm activity and in vivo toxicity.

### 3.5. Antibacterial Activity of M33_PEG 5000 NPs

On the basis of the promising results achieved with PEG5000, only M33_PEG5000 NPs underwent in vitro efficacy studies against bacteria. NPs produced with PEG5000 were tested at different final peptide concentrations (24 µM, 12 µM, 6 µM). Unloaded NPs were tested as a control. The activity was evaluated in terms of the growth of *P. aeruginosa* ATCC 27853™ 24, 48 and 72 h after treatment, compared with the control sample (untreated bacterial cells). Antibacterial activity persisted up to 72 h when M33_PEG5000 NPs were loaded at the higher concentrations, and up to 48 h with M33_PEG5000 NPs loaded with 6 µM SET-M33 (Figure 9). The persistent effect can be related to the sustained release of SET-M33 provided by NPs.

### 3.6. Antibiofilm Activity of Free SET-M33 and M33_PEG5000 NPs

Antibiofilm activity of M33_PEG5000 NPs was measured and compared to that of free SET-M33 peptide using *P. aeruginosa* (ATCC 27853™) as a bacterial model. The free peptide showed evident antibiofilm activity at all concentrations tested, proving to be strongest 72 h after the initial incubation (Figure 10). This could be due to the high stability of SET-M33 polymeric structure. The profile of antibiofilm activity of M33_PEG5000 NPs was very similar to that of the free peptide; it had a dose-dependent trend, and the greatest reduction in biomass was at 24 µM. Like the free form, M33_PEG5000 NP activity was strongest after 72 h. Of note, when M33_PEG5000 NPs were tested, only a reduced amount of SET-M33 was released and able to exert its antibiofilm activity. Nonetheless, the effect of M33_PEG5000 NPs appeared comparable to that observed when the total amount of SET-M33 was free. This effect can be related to the ability of NPs to release the peptide in a controlled manner over time.

### 3.7. Cytotoxicity of Free SET-M33 and M33_PEG5000 NPs

Since the NPs described here are intended to treat pneumonia via local delivery to the lungs, cytotoxicity was measured on human bronchial cells from a healthy individual (16HBE14o- cells) and a CF patient (CFBE41o- cells). M33_PEG5000 NPs and the free peptide were incubated for 48 h with cells to determine EC50. No toxicity was detected for M33_PEG5000 NPs, whereas the free peptide at the same concentrations as in the NPs showed an EC50 of 23 µM with 16HBE14o- cells (Figure 11A) and 21 µM with CFBE41o- cells (Figure 11B).

### 3.8. In Vivo Acute Toxicity of Free SET-M33 and M33_PEG5000 NPs

Mice (*n* = 5/group) were treated via i.t. administration of free SET-M33 and M33_PEG5000 NPs at the same peptide concentration (10 mg/Kg). A nebulization system reproducing aerosol application was used. Mice were checked for signs of toxicity for up to 96 h, using the following toxicity score: wiry coat and poor motility = mild signs; very wiry coat, abundant lachrymation and poor motility even under stimulation = manifest signs (Figure 12). No signs of toxicity were observed in mice treated via i.t. administration of M33_PEG5000 NPs and unloaded nanoparticles, confirming the absence of toxicity already seen in cytotoxicity tests, and the high biocompatibility of the material used. On the other hand, i.t. administration of free SET-M33 caused three deaths within 6 h, mild signs in one mouse and no signs in another mouse. The situations of the two surviving mice remained unchanged over the 96 h.

## 4. Discussion

As a local delivery route, pulmonary administration through nebulization systems is an ideal way to treat respiratory infections, inflammation and other pathological manifestations affecting the lungs. However, important determinants of the clinical outcomes of inhaled medicines are the concentration/persistency of the drug in the lungs, as well as its ability to overcome airway barriers. Optimized inhalable formulations should be able to deliver the drug intact to the lungs and to shield its interactions with the lung environment [37,38].

The peptide SET-M33 is under development to become a new antibacterial drug to treat lung infections, where traditional antibiotics often fail due to recurrent bacterial resistance and the presence of sticky mucus associated with infections and inflammation [39]. This aspect is especially critical in CF, where the submucosal glands and distal airways are obstructed by thick tenacious secretions. Indeed, CF mucus may strongly limit the drug concentration reaching the target (i.e., bacterial cells in the case of antimicrobial therapy). The route to the target is even more complicated for inhaled antimicrobial agents due to the biofilm-like mode of growth of certain bacterial species, such as *P. aeruginosa*, involved in lung infections.

A preclinical stage in the development of SET-M33, including toxicity and pharmacological safety assessments of the free peptide when administered by intravenous short infusion, has been concluded [40]. The toxicity results and efficacy data already reported [5,10,40] showed a satisfactory therapeutic index, even with the narrow range of doses used. Likewise, preliminary experiments with free SET-M33 administered intratracheally by nebulization systems gave promising results in terms of efficacy and toxicity, indicating a preliminary NOAEL in mice of about 4 mg/Kg/day. However, its clinical application would be greatly improved by a decrease in toxicity.

A possible way to decrease local toxicity, and consequently enlarge the dose range, is encapsulation of the peptide in biocompatible nanoparticles. We previously tested encapsulation systems with single-chain dextran nanoparticles [16]. Here, we developed PLGA nanoparticles with a PEG shell. By virtue of their biocompatibility and biodegradability, PLGAs are a promising class of materials for delivery of antimicrobial peptides to the lungs [18,19]. These biocompatible NPs can be administered to the lungs through nebulizers as aqueous dispersions or dry powder [18,41]. In our study, two different formulations of PEGylated PLGA-based NPs containing SET-M33 were prepared, using PEG of two different molecular weights (2000 or 5000 Da). Since PEG 5000 showed better performance in a translational perspective, SET-M33-containing NPs conjugated with PEG 5000 were chosen for further characterization in terms of antibacterial efficacy and toxicity.

Testing of this well-characterized nanosystem showed three important effects for lung delivery: (1) NPs containing SET-M33 passed biological barriers (lung mucus and bacterial alginate) while maintaining the local activity of the peptide; (2) the active molecule was released from NPs over an extended period of time, producing a prolonged antibacterial effect after a single administration; (3) SET-M33 toxicity was much reduced, while the molecule remained active in terms of antibacterial effect. Practically, the peptide contained within NPs did not provoke any detectable side effects in cells or in vivo at peptide doses in NPs that caused manifest toxicity when used as a free molecule.

In conclusion, the combination of polymeric NPs with a strong new antibacterial agent, such as the peptide SET-M33, produced an ideal antibacterial nanosystem for the development of a novel antibacterial agent. This nanosystem is attractive for the treatment of lung pathologies, where traditional antibiotics are losing their activity due to multidrug-resistant bacteria.

## Figures and Tables

**Figure 1 pharmaceutics-15-00003-f001:**
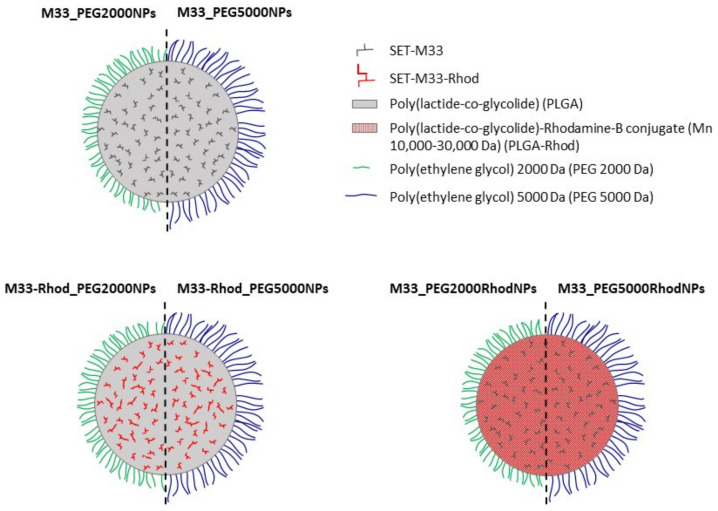
Diagram of the core–shell structure of SET-M33-loaded PLGA NPs, PLGA NPs containing SET-M33 conjugated with rhodamine (SET-M33_Rhod), and NPs prepared by employing PLGA conjugated with rhodamine (PLGA-Rhod).

**Figure 2 pharmaceutics-15-00003-f002:**
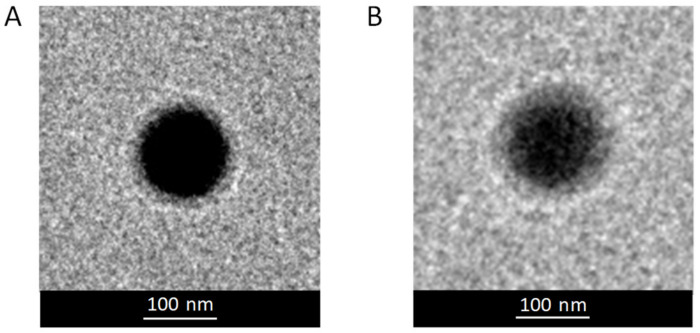
TEM images of M33-loaded NPs: M33_PEG2000 NPs (**A**) and M33_PEG5000 NPs (**B**). Images are representative of the samples.

**Figure 3 pharmaceutics-15-00003-f003:**
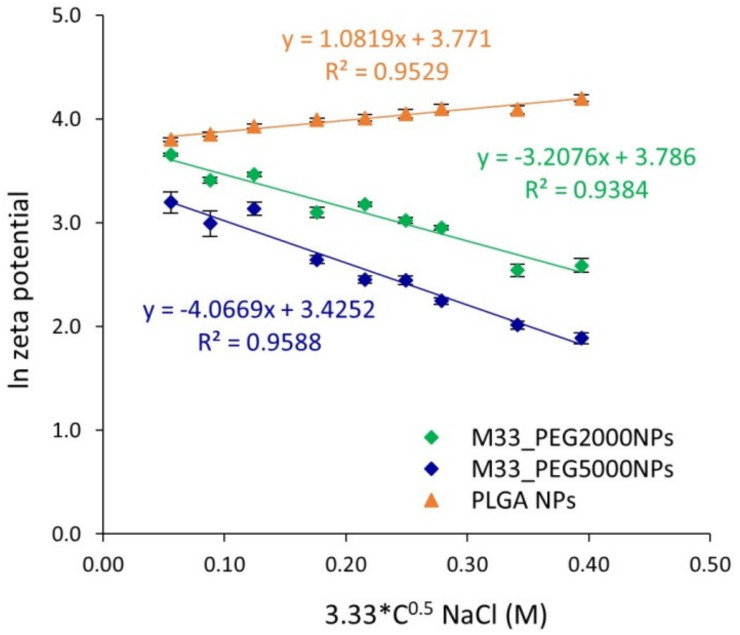
Fixed aqueous layer thickness (in nm) obtained as the slope of the linear regression of ionic strength of the dispersing medium against NP surface charge: ln (ζ) against k (k = 3.3 c0.5, where k − 1 is the Debye length). There is a significant difference between M33_PEG2000 NPs and M33_PEG5000 NPs (one-way ANOVA; *p* ˂ 0.05), and between both the two PEGylated formulations and the PLGA NP formulation control (one-way ANOVA; *p* ˂ 0.0001).

**Figure 4 pharmaceutics-15-00003-f004:**
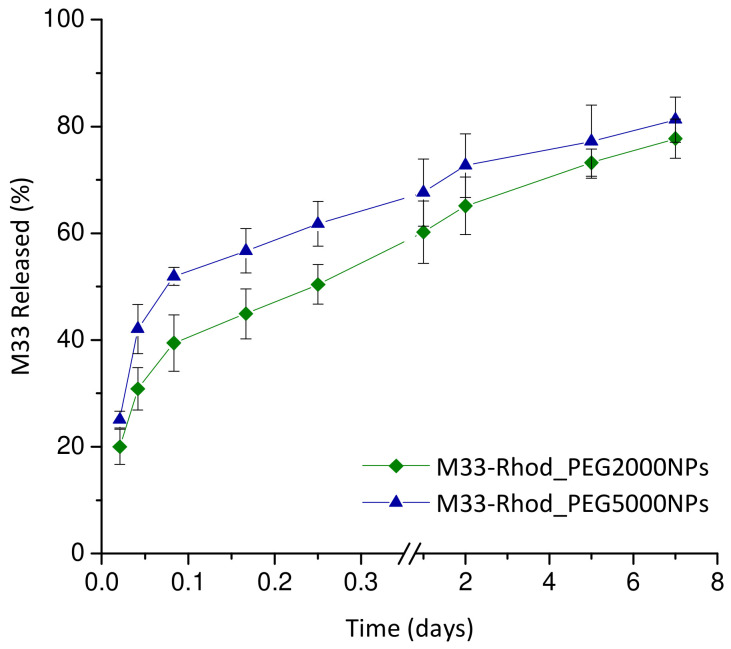
In vitro release of SET-M33_Rhod from PEG NPs calculated as the percentage of SET-M33 released by NPs (%) ± SD over time. The experiments were run in triplicate for each time point. No significant differences were observed (*t*-test; *p* > 0.05).

**Figure 5 pharmaceutics-15-00003-f005:**
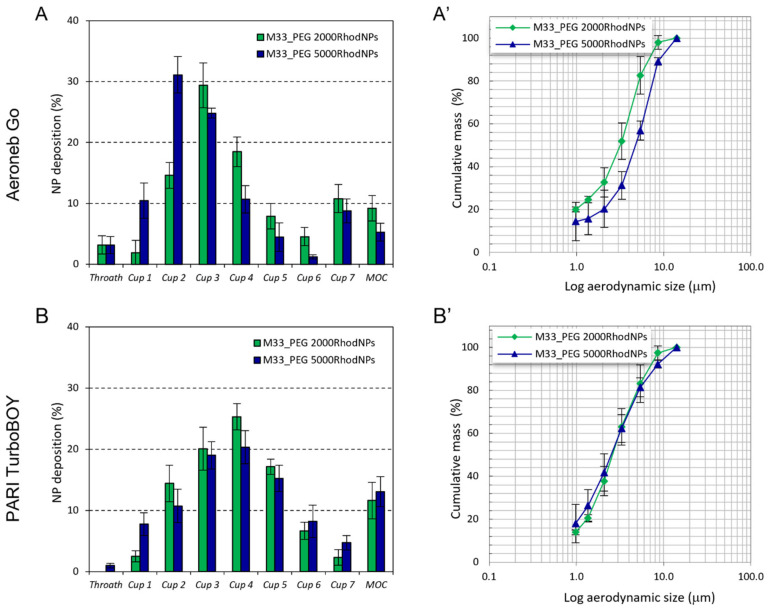
In vitro aerosol performance of M33_PEG 2000-Rhod and M33_PEG 5000-Rhod delivered through Aeroneb^®^ Go and PARI TurboBOY nebulizers: Cumulative mass recovered was a function of the cut-off diameter of the NGI stages (**A**,**B**), NGI deposition pattern (**A’**,**B’**). The emitted dose was measured as the difference between the total amount of NPs initially placed in the nebulizer chamber and the amount remaining. At the flow rate used (15 L/min), the cut-off diameters of the NGI stages were 0.98 µm (Cup 1), 1.36 µm (Cup 2), 2.08 µm (Cup 3), 3.3 µm (Cup 4), 5.39 µm (Cup 5), 8.61 µm (Cup 6), 14.1 µm (Cup 7). No significant differences were observed for all the formulations tested (one-way ANOVA multiple comparison; *p* > 0.05).

**Figure 6 pharmaceutics-15-00003-f006:**
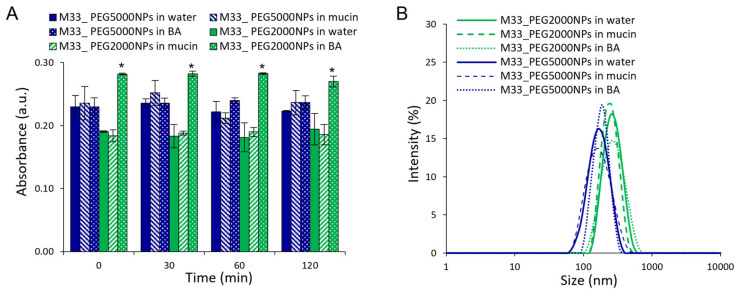
In vitro assessment of SET-M33-loaded NP interactions with mucin and BA: (**A**) Time trend of scattering at 650 nm by NPs (1 mg/mL) in the presence of mucin or BA. NP dispersions in water are reported as controls. (**B**) Size distribution of NPs with mucin or BA evaluated by dynamic light scattering intensity. The size distribution profiles of NP dispersions in water are reported as controls. The statistical analysis was performed by one-way ANOVA (* *p* ˂ 0.0001).

**Figure 7 pharmaceutics-15-00003-f007:**
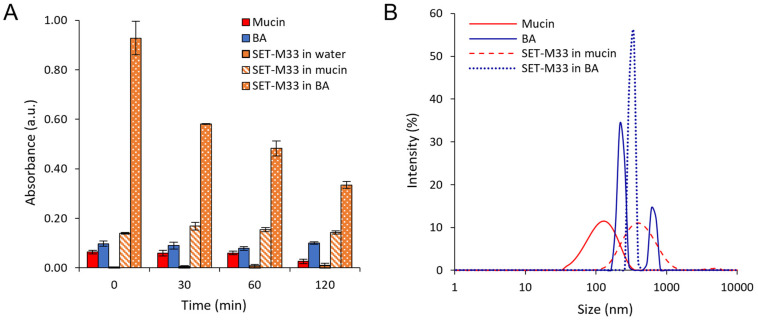
In vitro assessment of free SET-M33 interactions with mucin and BA: (**A**) Time trend of scattering at 650 nm by free SET-M33 (40 µg/mL) in the presence of mucin or BA. Scattering at 650 nm by SET-M33 solution in water is reported as the control. (**B**) Size distribution of free SET-M33 NPs with mucin or BA evaluated by dynamic light scattering intensity.

**Figure 8 pharmaceutics-15-00003-f008:**
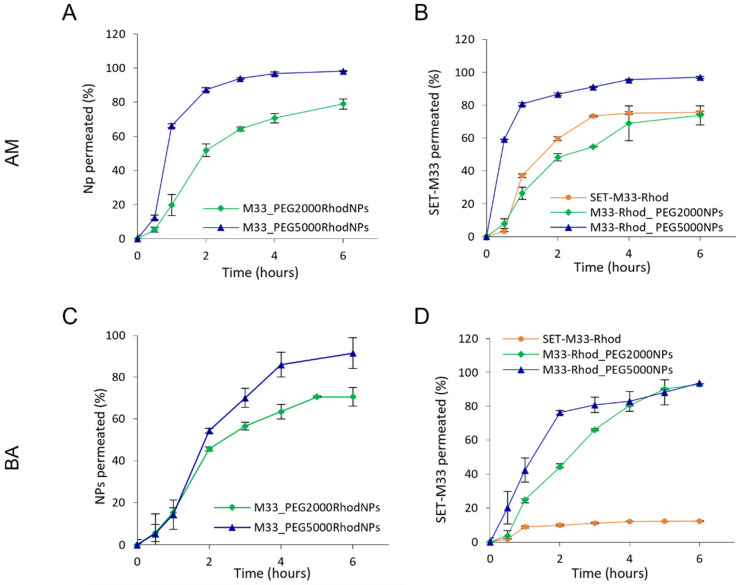
In vitro transport of SET-M33-loaded fluorescent NPs (**A**,**C**) and in vitro transport of SET-M33_Rhod-loaded NPs (**B**,**D**) through artificial mucus (AM) and bacterial alginates (BA), as determined by the Transwell^®^ multiplate assay. Results are presented as the percentage of SET-M33_Rhod and fluorescent NPs permeating across AM and BA as a function of time. Data are expressed as the mean ± SD calculated for three different batches (*n* = 6).

**Figure 9 pharmaceutics-15-00003-f009:**
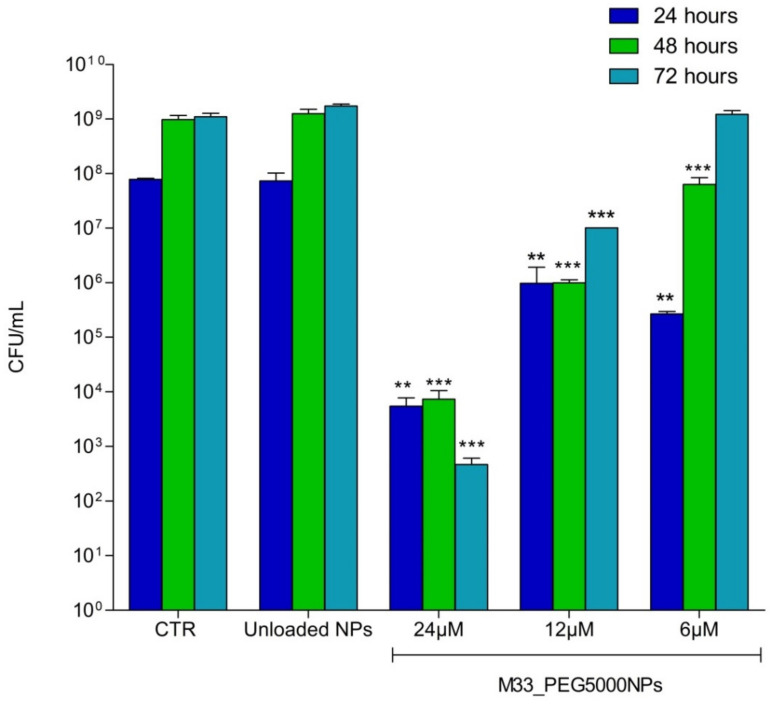
Histogram of *P. aeruginosa* (ATCC 27853™) colonies (CFUs) produced at different times (indicated in the internal legend) and then plated in LB agar medium: CTR indicates the number of colonies produced by untreated bacteria. Unloaded NPs indicates the number of colonies produced by bacteria incubated with NPs without peptide. M33_PEG5000 NPs show the number of colonies produced by bacteria incubated with NPs with the peptide at the concentrations indicated. The experiment was performed in triplicate. Results are expressed as mean ± SD of three wells. Significant differences between groups were evaluated by one-way ANOVA with Dunnett’s multiple comparison test using GraphPad Prism 5.03 software, where ***, *p* < 0.001; **, *p* < 0.05. Each column is compared with its control at each time point.

**Figure 10 pharmaceutics-15-00003-f010:**
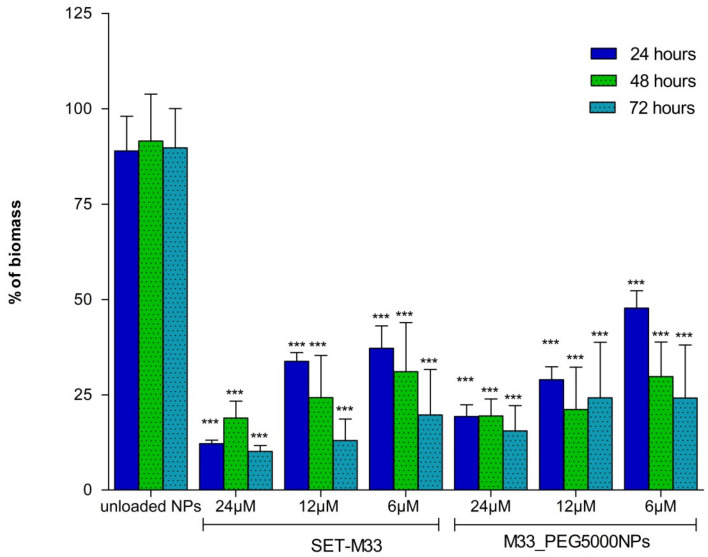
Histogram of the percentage of *P. aeruginosa* (ATCC 27853™) biofilm biomass at different times of incubation with free SET-M33 or SET-M33 peptide encapsulated in nanoparticles (M33_PEG5000 NPs): Unloaded NPs indicates the biofilm biomass after incubation with nanoparticles without peptide. Results are expressed as mean ± SD of three wells. Significant differences between groups were evaluated by one-way ANOVA with Dunnett’s multiple comparison test using GraphPad Prism 5.03 software, where ***, *p* < 0.001. Each column is compared with its control at each time point.

**Figure 11 pharmaceutics-15-00003-f011:**
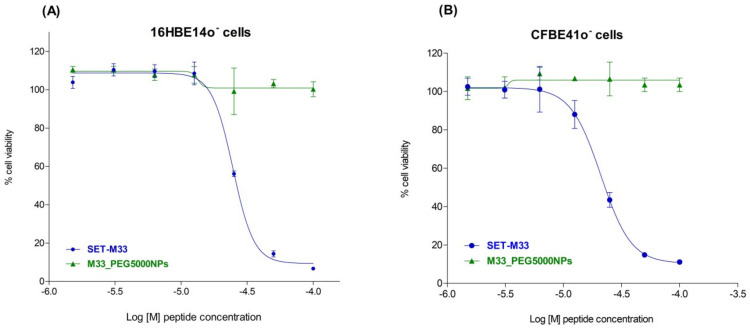
Cytotoxicity of free SET-M33 (circle) and M33_PEG5000 NPs (triangle) towards 16HBE14o- (**A**) and CFBE41o- (**B**) bronchial cells: The cell lines were incubated with free and encapsulated peptide at the same concentrations (from 100 to 3.125 µM) for 48 h. Cell viability was then evaluated using 0.1% crystal violet staining solution. Cell survival (*y*-axis), expressed as a percentage with respect to untreated cells, is plotted against peptide concentration (*x*-axis) on a logarithmic scale. The experiment was performed in triplicate. The data, reported as mean ± SD (*n* = 3), were analyzed by non-linear regression using GraphPad Prism 5.03.

**Figure 12 pharmaceutics-15-00003-f012:**
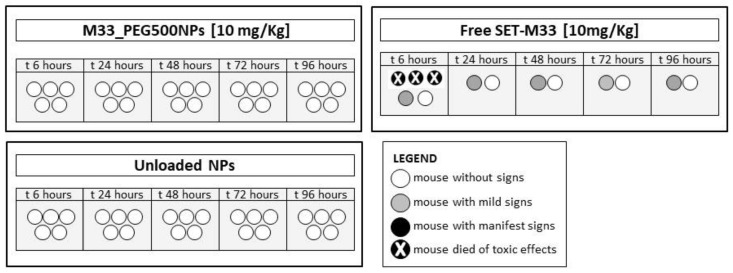
Acute toxicity of M33_PEG5000 NPs, free SET-M33 and unloaded NPs in vivo: Mice (represented as circles) underwent i.t. inoculation with 10 mg/kg in a single dose and were monitored for 96 h. Different scales of grey and the X symbols in the circles indicate severity of signs and death, as described in the internal legend.

**Table 1 pharmaceutics-15-00003-t001:** Overall properties (hydrodynamic diameter (DH), polydispersity index (PDI), ζ potential, encapsulation efficiency (EE) and actual loading (AL)) of SET-M33-loaded PLGA NPs.

Formulation	D_H_ (nm) ± SD	PDI (mean) ± SD	ζ potential (mV) ± SD	EE (%) ±SD	AL (mg SET-M33/100 mg NPs) ± SD
M33_PEG2000 NPs	208.8 ± 1.5	0.141 ± 0.020	−18.4 ± 0.7	93.9 ± 1.25	3.793 ± 0.045
M33_PEG5000 NPs	185.7 ± 1.8	0.09 ± 0.044	−18.87 ± 0.6	85.9 ± 4.63	3.557 ± 0.159

D_H_, hydrodynamic diameter; PDI, polydispersity index; EE, encapsulation efficiency, calculated as actual loading/theoretical loading × 100; AL, actual loading.

**Table 2 pharmaceutics-15-00003-t002:** Overall aerosol properties of M33_PEG 2000-Rhod and M33_PEG 5000-Rhod delivered through Aeroneb^®^ Go and PARI TurboBOY nebulizers: Experimental mass median aerodynamic diameter (MMADexp) and geometric standard deviation (GSD) were calculated by plotting cumulative mass of particles retained in each collection cup versus the cut-off diameter of the corresponding stage. The fine particle fraction (FPF) was calculated as the percentage of NPs deposited in stages 3–7 (MMADexp < 5.39 μm) compared to the amount initially loaded in the nebulizer chamber. The respirable fraction (RF) is the total amount recovered from the NGI. Data are reported as mean ± SD (*n* = 3).

	M33_PEG 2000RhodNPs		M33_PEG 5000RhodNPs	
	Aeroneb	PARI TurboBOY	Aeroneb	PARI TurboBOY
FPF	37.1 ± 16.3	39.4 ± 9.5	40.5 ± 0.1	43.0 ± 5.7
RF	71.1 ± 1.6	71.2 ± 7.4	50.0 ± 7.8	67.5 ± 2.2
MMAD_exp_	2.68 ± 1.03	2.66 ± 1.02	3.85 ± 0.70	2.53 ± 0.71
GSD	2.91 ± 0.98	2.66 ± 0.58	2.62 ± 0.02	2.86 ± 0.47

FPF, fine particle fraction; RF, respirable fraction; MMADexp, experimental mass median aerodynamic diameter; GSD, geometric standard deviation.

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
