# Peer review of "Inhalable Polymeric Nanoparticles for Pulmonary Delivery of Antimicrobial Peptide SET-M33: Antibacterial Activity and Toxicity In Vitro and In Vivo"

_pharmaceutics, 2022, doi:10.3390/pharmaceutics15010003_

Round 1

Reviewer 1 Report

1.  lines 340-440 stats need to be done and P-Value should be reported.

2. If no statical difference is found between 2 engineered NPs when employing PLGA conjugated with PEG 2000 Da or 5000 Da? How would you explain the changes in particle size when using the same methodology? 

3. lines 338 how did you calculate the EE= Encapsulation efficiency?

4. line 352 which kinetic model would both NPs formulations release follow?

5. How would your deal with NP local lung accumulations, toxicity, and the delay in drug release over the long-term treatment? especially in such bacterial infection conditions.

6. lines 358-360, it is clear that when particle size is low the loaded drug is released faster, where is the stat?  clarify the results.

7.  What's the effect of polymer size and type on particle size?

8. In Figure 4. why there is a significant change in RF for M33_PEG 5000-RhodNPs?

9. lines 432-434,"the highest diffusion was observed for NPs modified with PEG 5000 Da,"  Although increasing the molecular weight can result in decreasing permeability. You may need to explain this more. 

10. legends in figure 6 are not clear to me, and 6B needs slight adjustments.

11. caption in Figure 7, adjust to the bottom.  

12. in Figure 10 and Figure 11, did you compare the same amount of drugs free and encapsulated or you are comparing the free drug amount to the total NP weight i.e 10mg/kg free drug against 10 mg of total NP? Explain this in both figures.

12. discussion may need   improvement and more focus on the study findings

Author Response

  1. lines 340-440 stats need to be done and P-Value should be reported.

The statistical analysis was added in the new version of the manuscript.

  1. If no statical difference is found between 2 engineered NPs when employing PLGA conjugated with PEG 2000 Da or 5000 Da? How would you explain the changes in particle size when using the same methodology? 

Any interaction between the peptide SET-M33 and the polymeric matrix cannot be excluded. The presence of PEG in the polymeric matrix can reduce the interactions between SET-M33 and the polymeric matrix and provide the formation of nanometric systems. Employing PEG 5000 Da Mw, a higher stabilization effect (lower PDI) and lower particle size than that observed for PEG 2000 Da modified NPs was observed. Nevertheless, the differences observed in the two pegylated NPs are very slight and both the formulations showed optimal properties for the pulmonary delivery.

  1. lines 338 how did you calculate the EE= Encapsulation efficiency?

The encapsulation efficiency was calculated as actual loading/theoretical loading x 100, as reported in the methods section (row 161-162). This information was added also in the result section, at row 34-341, in the new version of the manuscript.

  1. line 352 which kinetic model would both NPs formulations release follow?

The solubility of the encapsulated drug, its diffusion, and the biodegradation of the polymeric matrix govern the release of a drug embedded into a polymeric matrix. In the case of polymeric nanoparticles, the diffusion of the drug is generally faster than polymeric matrix erosion, thus the mechanism of release is largely controlled by a diffusion process, according to the mathematical Higuchi model (correlation coefficient higher than 0.9 for both the tested formulations). A comment on the SET-M33 release profile was added in the result section.

  1. How would your deal with NP local lung accumulations, toxicity, and the delay in drug release over the long-term treatment? especially in such bacterial infection conditions.

In our previous article, we demonstrated that there was no significant change in the expression of inflammation-associated genes, such as IL-6, IL-10 and the tumour necrosis factor-α TNF-α and NF-κB, in the lungs at 36 h after pulmonary administration in mice of PLGA-based NPs, suggesting a safe use of NPs for delivery at lungs. These findings were also supported by the invariant production of the proinflammatory cytokine IL-6 at the protein level, in the BAL of mice at 36 h after i.t. administration of PLGA NPs, compared with healthy mice receiving PBS as the vehicle control (Casciaro B et al Biomacromolecules, 2019, 13;20(5):1876-1888). More recently, we evaluated the effect of PLGA NPs on the global genetic profile of lungs upon administration in the conductive airways of healthy mice, in comparison to the vehicle-treated animals, and interestingly, only six genes out of 25,000 were found to be up/downregulated in animals treated with NPs versus PBS-treated animal, and none of these genes appear to be directly related to toxicity-related processes (Cappiello et al, Pharmaceutics, in press). The lung toxicity and accumulation of polymeric nanoparticles after pulmonary delivery is still an open issue and numerous studies are still needed to clarify this aspect and fill the gap between research and clinics.

  1. lines 358-360, it is clear that when particle size is low the loaded drug is released faster, where is the stat?  clarify the results.

As suggested by the referee, the slight difference in the release profile can be due to the low differences observed in the NP size. Nevertheless, according to the statistical analysis, no significant differences were observed (p>0.05) in the SET-M33 release kinetics from the two tested formulations. The discussion of release data was improved in the new version of the manuscript.

  1. What's the effect of polymer size and type on particle size?

As reported in the previous answer, the encapsulation of SET-M33 strongly affected the nanoparticles properties, in fact NPs containing the peptide, prepared without PEG appeared very big (size higher than 500 nm), unstable and not suitable for the pulmonary delivery, suggesting that interactions between the peptide SET-M33 and the polymeric matrix are present. The use of PEGylated PLGA matrix can reduce the interactions between SET-M33 and the polymeric matrix and provide the formation of stable nanometric systems with properties, i.e. size and surface charge, suitable for the pulmonary delivery of the drug. The observed slight differences in the particle size between the PEG5000 and PEG2000 engineered NPs can be related to the ability of PEG with higher molecular weight to better reduce the interactions between the SET-M33 and the polymeric matrix. Nevertheless, the differences observed in the two pegylated NPs are very slight and both the formulations showed optimal properties for the pulmonary delivery.

  1. In Figure 4. why there is a significant change in RF for M33_PEG 5000-RhodNPs?

The aerosolization performance, and thus the RF, of a pharmaceutical formulation, predominantly depends on the nebulization device and formulation properties. Between air-jet and vibrating mesh nebulizers there is a difference between the shear forces exerted on the formulation (air-jet nebulizer, i.e. Pari Turbo boy, exerts higher stress shearing forces than mesh nebulizer, i.e. Aeroneb). The shearing forces given during nebulization can lead to particle aggregation deformation and/or disassembling, especially when lipid or natural polymer are used. Nevertheless, recent studies underline that the presence of a dense and optimized PEG shell can improves the pulmonary delivery of nanocarriers following nebulization (Kim et al ACS Nano 2022, 16, 14792−14806; Lokugamage et al Nat Biomed Eng . 2021 Sep;5(9):1059-1068). Thus, many aspects influence the nebulization process of nanometric systems and the correlations between them is a very complex issue. Although our results suggest the influence of PEG molecular weight, it is hard to speculate on its effect on NP nebulization behaviour on the basis of presented results. A new study should be designed in order to enlarge upon this aspect.

  1. lines 432-434,"the highest diffusion was observed for NPs modified with PEG 5000 Da,"  Although increasing the molecular weight can result in decreasing permeability. You may need to explain this more.

The higher diffusion across the mucus layer observed for the PEG5000 modified formulations versus PEG2000 can be related to the PEG shell with higher thickness, as demonstrated by FALT assay, and density. Although the two PEGylated NPs diffused rapidly through mucus layer, the best mobility of NPs and diffusion across the mucus layer is observed when PEG shell is characterized by an optimal density and conformation (brush), which is generally provided by PEG with a molecular weight of 5000 Da (Xu Q et al J Control Release, 2013, 170; 279-286).

10. legends in figure 6 are not clear to me, and 6B needs slight adjustments.

11. caption in Figure 7, adjust to the bottom.

We apologize with the reviewer for the wrong format. The correct image is now reported in the new version of the manuscript.

12.     in Figure 10 and Figure 11, did you compare the same amount of drugs free and encapsulated or you are comparing the free drug amount to the total NP weight i.e 10mg/kg free drug against 10 mg of total NP? Explain this in both figures.

Of course, the comparison was done using the same concentration of peptide, free or encapsulated in NPs. In the caption of figure 10 the following sentence is already present: ‘The cell lines were incubated with free and encapsulated peptide at the same concentrations’.  In the text of in vivo toxicity (regarding Fig. 11) the following sentence is already present: ‘Mice (5/group) were treated i.t. with free SET-M33 and M33_PEG5000 NPs at the same peptide concentration (10 mg/Kg)’.

13.  discussion may need improvement and more focus on the study findings

We thank the reviewer for the suggestion. The new version of the manuscript presented an improved discussion of the presented results.

Reviewer 2 Report

In the present manuscript, Laura Cresti et al. reported the pulmonary delivery of antimicrobial peptide SET-M33. The authors used pegylated PLGA for encapsulating the antimicrobial peptide in inhalable nanoparticles.  Nanoparticle formulations were comprehensively evaluated in vitro release and in vivo fate.  These findings are interesting, and I recommend this manuscript for publication in Pharmaceutics after addressing the following concern:

1.      Characterization data of SET-M33 peptide and rhodamine conjugates is not provided in the manuscript.

2.      Throughout the in vitro characterization and evaluation study, control PLGA (without PEG) nanoparticles (with and without peptide) were not included. Even for predicting the core-shell arrangement of PEG-PLGA nanoparticles. The control PLGA nanoparticles data set is highly desired and should be included.

3.      Figure 5 A: why absorbance values not changed with time when nanoparticles were incubated in water. These results should be elaborated on.

4.      In figure 6 few legends are missing. In a few other figures as well, the legends are not consistent.

5.      Lower AM (artificial CF mucus) permeation of M33-PEG2000RhodNPs than SET-M33-Rhod needs further elaboration.

6.      A correlation between in vitro release and time-dependent in vitro efficacy should be discussed.

7.      How in vivo acute toxicity dose was decided without knowing the biodistribution and in vivo efficacy dose? A detailed histological evaluation of nanoparticle distribution and toxicity should have been performed.

Author Response

In the present manuscript, Laura Cresti et al. reported the pulmonary delivery of antimicrobial peptide SET-M33. The authors used pegylated PLGA for encapsulating the antimicrobial peptide in inhalable nanoparticles.  Nanoparticle formulations were comprehensively evaluated in vitro release and in vivo fate.  These findings are interesting, and I recommend this manuscript for publication in Pharmaceutics after addressing the following concern:

  1. Characterization data of SET-M33 peptide and rhodamine conjugates is not provided in the manuscript.

We do not perfectly understand what the reviewer means. Characterization of peptide-rhodamine conjugate is fully described in material and methods (lines 107-108 and 119-122), and in the results (lines 352-354, Fig 3, 378-381, Fig 4).

  1. Throughout the in vitro characterization and evaluation study, control PLGA (without PEG) nanoparticles (with and without peptide) were not included. Even for predicting the core-shell arrangement of PEG-PLGA nanoparticles. The control PLGA nanoparticles data set is highly desired and should be included.

The encapsulation of SET-M33 strongly affected the nanoparticles properties, in fact, NPs containing the peptide, prepared without PEG appeared very big (size higher than 600 nm), unstable and not suitable for the pulmonary delivery. Thus, the comparison between the PEGylated and non-PEGylated SET-M33-loaded NPs cannot be considered representative. Concerning the PEG shell characterization, in the FALT experiment we have reported data concerning the unloaded PLGA NPs, that demonstrate that the addiction of PEG to the formulation lead to the shell coating formation, not detectable in non-PEGylated NPs. Furthermore, the thickness of the fixed aqueous layer is increased by increasing the PEG molecular weight.

3. Figure 5 A: why absorbance values not changed with time when nanoparticles were incubated in water. These results should be elaborated on.

Generally, NP aggregation or the increase in particle size can lead to a change in absorbance. The scattering of all the tested formulation in water did not change during the time demonstrating the stability of the tested formulations during the time.

  1. In figure 6 few legends are missing. In a few other figures as well, the legends are not consistent.

We apology with the referee for the incorrect format of the manuscript. In the new version of the manuscript the images and legends were reported correctly.

  1. Lower AM (artificial CF mucus) permeation of M33-PEG2000RhodNPs than SET-M33-Rhod needs further elaboration.

The diffusion ability is strongly affected by the size/molecular weight of the diffusing molecule. It is generally recognized that large molecules diffuse slowly. In absence of interaction between the drug and the diffusion medium, the free SET-M33, can diffuse faster than the encapsulated form, characterized by a nanometric size. Furthermore, the permeation profile f NPs (PEGRhodNPs) is very close to that observed for encapsulated SET-M33, suggesting that, in this conditions, the release of SET-M33 from NPs is not relevant and all the SET-M33 detected into the acceptor compartment is in the encapsulated form.

  1. A correlation between in vitro release and time-dependent in vitro efficacy should be discussed.

The profile of anti-biofilm activity of M33_PEG5000 NPs was very similar to that of the free peptide; it had a dose-dependent trend and the greatest reduction in biomass was at 24 µM. Like the free form, M33_PEG5000 NP activity was strongest after 72 hours. Of note, when M33_PEG5000 NPs were tested only a reduced amount of SET-M33 is released and able to exert its antibiofilm activity. Nonetheless, the effect of M33_PEG5000 NPs appeared comparable to that observed when the total amount of SET-M33 is free. This effect can be related to the ability of NPs to release the peptide in a controlled manner during the time.

  1. How in vivo acute toxicity dose was decided without knowing the biodistribution and in vivo efficacy dose? A detailed histological evaluation of nanoparticle distribution and toxicity should have been performed.

The peptide SET-M33 is studied since many years as a possible new candidate for the development of a novel antimicrobial drug. For a full description of this molecule in terms of efficacy (sepsis, lung infections, skin infections), toxicity, bio-distribution, excretion, selection of resistances, gene toxicity, mechanism of action, immunomodulatory activity, time-kill concentrations etc etc., please, see the following articles that you can found cited also in the present paper: Pini et al., 2010, FASEB J; Brunetti et al., 2016, Sci Reports; Brunetti et al., 2016, J Biol Chem; van Der Weide et al., 2017, BBA Biomembranes; van der Weide et al., 2019; Int J Antimicrob Agents. A new paper about safety evaluations of SET-M33 in rats and dogs, with a full characterization about Dose Range Finding, NOAEL, neurological, respiratory and cardiology toxicity, pharmacokinetics parameters, along with histological evaluations, has been just published (Cresti et al., 2022, Sci Reports 12:19294). This latter article has been added as ref 40 to the present paper.